# SegMaST: Mamba-based Spatio-Temporal Modeling to Improve Longitudinal Disease Detection and Segmentation

**Aswathi Varma**[1,2,3]      ASWATHI@TUM.DE
**Jonas Weidner**[1,3]      J.WEIDNER@TUM.DE
**Laurin Lux**[2,3]      LAURIN.LUX@TUM.DE
**Cosmin Bercea**[3,4,5]      COSMIN.BERCEA@TUM.DE
**Mark Mühlau**[6]      MARK.MUEHLAU@TUM.DE
**Jan Kirschke**[7]      JAN.KIRSCHKE@TUM.DE
**Benedikt Wiestler**[*,1,3]      B.WIESTLER@TUM.DE
**Daniel Rueckert**[*,2,3,8]      DANIEL.RUECKERT@TUM.DE

[1] *AI for Image-Guided Diagnosis and Therapy, TUM, Munich, Germany*

[2] *Chair for AI in Healthcare and Medicine, Technical University of Munich (TUM) and TUM University Hospital, Munich, Germany*

[3] *Munich Center for Machine Learning, Munich, Germany*

[4] *Technical University of Munich, Munich, Germany*

[5] *Helmholtz AI and Helmholtz Center Munich, Munich, Germany*

[6] *Department of Neurology, TUM University Hospital, Munich, Germany*

[7] *Department of Neuroradiology, TUM University Hospital, Munich, Germany*

[8] *Department of Computing, Imperial College London, London, UK*

**Editors:** Accepted for publication at MIDL 2026

## Abstract

Longitudinal medical image segmentation is fundamental for quantifying disease progression and evaluating treatment efficacy. However, two critical challenges persist: First, methods that jointly segment baseline and follow-up images remain underexplored, often missing the contextual benefits of simultaneous assessment and lacking longitudinal consistency. Second, real-world datasets typically exhibit severe class imbalance, as scans showing actual disease progression are far rarer than those showing stable anatomy, an issue frequently neglected by existing models. To address these limitations, we propose *SegMaST*, a novel *Mamba*-based spatio-temporal framework. Unlike conventional approaches that treat timepoints in isolation, *SegMaST* leverages cross-temporal information and spatial correspondences to jointly segment the initial baseline mask and explicitly localize new or progressive pathologies in follow-up scans. Additionally, we introduce an imbalance-aware loss accumulation strategy to enhance robustness in realistic clinical settings. On longitudinal cohorts of patients with Multiple Sclerosis (MS) and glioma, *SegMaST* outperforms established CNN- and attention-based baselines for follow-up segmentation (mean follow-up Dice MS *in-house* 0.536, *MSSEG-2* 0.620, and glioma 0.631) and lesion detection (F1 *in-house* 0.688, *MSSEG-2* 0.723), while maintaining state-of-the-art accuracy in baseline segmentation (Dice: 0.617 MS, 0.844 glioma).

**Keywords:** Longitudinal Segmentation, State Space Models, Mamba, Imbalance Aware Loss

---

* Contributed equally as senior authors

## 1. Introduction

Longitudinal Magnetic Resonance Imaging (MRI) is the cornerstone of disease assessment for neurological pathologies. Unlike single-timepoint analysis, longitudinal imaging captures disease progression and therapy response over time (Martí-Juan et al., 2020). This is particularly critical in *Multiple Sclerosis* (MS), where the detection of new and evolving lesions serves as the primary biomarker for modifying treatment plans. Similarly, in *diffuse glioma*, precise follow-up segmentation is required to distinguish tumor recurrence from treatment effects. While manual volumetry remains the gold standard (Barkhof et al., 2025; Wen et al., 2023), it is labor-intensive and prone to inter- and intra-variability. Deep learning offers a scalable solution to this bottleneck; however, popular architectures like *UNet* (Ronneberger et al., 2015) and *Swin-UNETR* (Hatamizadeh et al., 2021) typically process time-points independently, ignoring the rich temporal correlations in serial imaging. This risks *longitudinal inconsistency*, where lesions appear and disappear between scans due to noise, whereas deformable registration introduces alignment errors that obscure progression.

To ensure longitudinal consistency, recent architectures explicitly model temporal dependencies. Simple input concatenation, as in *Neuropoly* (Macar et al., 2021), often fails to separate static anatomy from true temporal change. In contrast, *SNAC* (Cabezas et al., 2021) uses parallel encoders to compare features across multiple resolutions, offering clearer temporal cues. Advanced methods explicitly model scan differences to reduce longitudinal inconsistency: *CoActSeg* (Wu et al., 2023) uses voxel-wise difference maps, and the *Temporal Difference Weighting (TDW)* block (Rokuss et al., 2024) performs subtraction on latent features to amplify evolving regions.

Architectural design is often shaped by the available annotation scheme. One common strategy fully annotates both time points, enabling temporally consistent learning (Carass et al., 2017; Wei et al., 2021) but at the cost of extensive manual labelling. Another approach reduces this burden by annotating only new or progressing abnormalities at follow-up (Commowick et al., 2021), though models trained under this setting (Macar et al., 2021; Cabezas et al., 2021) often lack the anatomical context needed to interpret changes. A third annotation configuration, followed in our work, strikes a balance by providing a full baseline segmentation while annotating only new or enlarging regions in the follow-up scans. However, prior efforts often suffer from significant limitations. Some rely on cumbersome auxiliary inputs (Doga Basaran et al., 2024), while others require labor-intensive, full-mask supervision for every longitudinal scan (Denner et al., 2020). Consequently, the hybrid setting remains underexplored.

Furthermore, typically a strong class imbalance of stable *vs.* progressive cases exists in real-world longitudinal data. This imbalance drives models toward two extremes: majority-class bias (missing subtle progression) or over-sensitivity (falsely flagging stable patients) - both challenge the safe clinical use of such models. To mitigate this issue, strategies range from augmenting training sets with synthetic abnormalities (Tahghighi et al., 2024) to leveraging abundant cross-sectional data to compensate for limited longitudinal pairs (Wu et al., 2023). However, ensuring robustness across both progressing and non-progressing cases remains an open problem. Additionally, enhancing baseline segmentation alongside progression detection ensures more reliable longitudinal analysis. To this end, our contributions are as follows:

1. We introduce *SegMaST*, a *Mamba*-based spatio-temporal framework that exploits longitudinal dependencies through efficient state-space modeling to simultaneously refine baseline anatomical segmentation and precisely localize new or enlarging abnormalities in follow-up scans.

2. We address the clinically relevant issue that many scans show no disease activity at follow-up, by employing an *imbalance-aware loss accumulation*. This ensures robust performance on real-world clinical data where disease activity is intermittent.

3. We extensively validate *SegMaST* on two distinct pathologies (MS and glioma), demonstrating that it significantly reduces longitudinal inconsistency compared to both cross-sectional and longitudinal baselines, effectively distinguishing true disease progression from noise in stable and active cases.

## 2. Methodology

The above contributions motivate a methodological design that combines longitudinal feature modeling with stable learning in non-progressing cases, along with a joint prediction strategy for segmenting baseline regions and identifying follow-up abnormalities. Following this design, *SegMaST* processes baseline and follow-up scans within one coherent pipeline to generate both masks (see Figure 1-(a)). To balance performance and efficiency, we adopt a 2.5D architecture rather than a resource-intensive full-3D model, preserving contextual information while significantly lowering resource demands.

To effectively capture nuanced spatio-temporal interactions within this framework, our pipeline leverages *Mamba*-based state-space modeling. This approach addresses the computational bottlenecks inherent in standard spatial self-attention, which suffers from quadratic complexity ($O(N^2)$) in terms of sequence length. By strictly utilizing a selective state-space model (SSM) with linear complexity ($O(N)$) (Gu and Dao, 2023), we overcome the challenges that arise when flattening spatial patches results in sequence lengths that are prohibitive for explicit attention matrices. Consequently, we achieve global receptive fields comparable to *SegFormer* (Xie et al., 2021) while maintaining high throughput. We realize this design through a modular pipeline consisting of a hierarchical encoder, where Spatio-Temporal (ST) blocks capture longitudinal disease activity at multiple scales, and a dual-head decoder, detailed as follows.

**Hierarchical Encoder.** To capture features ranging from fine-grained lesion boundaries to global semantic context, *SegMaST* employs a four-layer hierarchical encoder. For an input pair with spatial dimensions $H \times W$ and channels $C$, we define the input tensor as $X \in \mathbb{R}^{N \times C \times H \times W}$, where $N = 2$ represents the timepoints. We first apply a convolutional patch embedding layer to tokenize the input while preserving local spatial continuity, yielding $X_1 \in \mathbb{R}^{N \times C_1 \times \frac{H}{4} \times \frac{W}{4}}$. In subsequent stages, we progressively increase the receptive field, such that the feature tensor at layer $i$ follows $X_i \in \mathbb{R}^{N \times C_i \times \frac{H}{2^{i+1}} \times \frac{W}{2^{i+1}}}$.

**ST Block and MaST Module.** We employ ST blocks to capture joint dynamics from the encoder features. Central to this block is the *Mamba* Spatio-Temporal (*MaST*) module, which processes the input by layer normalizing and reshaping it into two complementary sequences: (1) **Temporal-first** ($X_{i,t} \in \mathbb{R}^{C_i \times N(H/2^{i+1}W/2^{i+1})}$), where spatial patches are un-

folded and concatenated along the time axis; (2) **Spatial-first** ($X_{i,s} \in \mathbb{R}^{C_i \times (H/2^{i+1} W/2^{i+1})N}$), where patches are stacked to preserve spatial correspondence across time. These flattened sequences, which create contexts too long for standard attention, are efficiently processed by the *Mamba* SSM (Gu and Dao, 2023). By compressing context into a hidden state rather than calculating pairwise interactions, the SSM enables the effective learning of spatio-temporal dependencies with linear complexity. We repeat this ST block $M = 4$ times before applying overlapped patch merging to downsample features for the next stage.

**Decoder.** We utilize a *CNN*-based decoder to generate segmentation masks, adopting the lightweight design of (Yang et al., 2024; Xie et al., 2021). First, an *MLP* unifies the multi-level features from the encoder along the channel dimension. These unified representations are upsampled to a common spatial resolution and concatenated. A subsequent *MLP* fuses this aggregated representation, projecting the concatenated tensor $(N, 4C, H/4, W/4)$ to a lower embedding dimension $C$. The resulting features are then reshaped to recover the temporal structure, separating the latent representations of the two time points. Finally, a dual-head prediction module applies a $1 \times 1$ convolution to the respective time steps, producing the baseline segmentation $\hat{Y_1}$ from the first time point and the progression segmentation $\hat{Y_P}$ from the second time point.

**Imbalance-Aware Loss Accumulation.** In clinical settings, follow-up datasets are dominated by stable cases. These cases have *empty* progression masks. Progressive cases, which contain *non-empty* masks, are comparatively rare. This imbalance skews model predictions and reduces sensitivity to disease evolution (Karimian-Jazi et al., 2018). We address this through a combination of an *imbalance-aware* loss accumulation that filters some empty masks and a sampling strategy that maintains a balanced progression-to-no-progression ratio per batch. This prevents gradients from being dominated by empty masks while preserving useful learning signals.

Given a batch of segmentation outputs, let $Y_1, \hat{Y}_1$ denote the ground truth and predicted baseline masks, and $Y_P, \hat{Y}_P$ denote the ground truth and predicted progression masks. We use the *Dice-Focal* loss, computed separately for the baseline and progression heads, and sum them to obtain the final loss. To prevent trivial cases from dominating training, we apply a filtering mechanism for progression head loss, as many samples have no progression ($Y_P = 0$).

**Filtering Mechanism.** The total loss consists of two terms: the baseline head loss ($\mathcal{L}_1$) and a subset of the progression head loss ($\mathcal{L}_p$). We define a binary indicator $M$, indicating whether a sample contains progression. Using this value, we separate the progression loss values into:

$$\mathcal{L}_p^+ = \mathcal{L}_p[M], \quad \mathcal{L}_p^0 = \mathcal{L}_p[\neg M]$$

Here $\mathcal{L}_p^+$ represents the loss for samples with progression ($M = 1$), while $\mathcal{L}_p^0$ corresponds to the loss for no-progression samples ($M = 0$). Since we want the model to learn from no-progression cases while preventing imbalance, we randomly sample a subset from $\mathcal{L}_p^0$, matching the number of progression cases in the batch. This selected subset is included in the final loss term, denoted as $\mathcal{L}_p^\emptyset$. The filtered progression loss is obtained by concatenating the two:

$$\mathcal{L}_p^{\text{filtered}} = Concat(\mathcal{L}_p^+, \mathcal{L}_p^\emptyset).$$

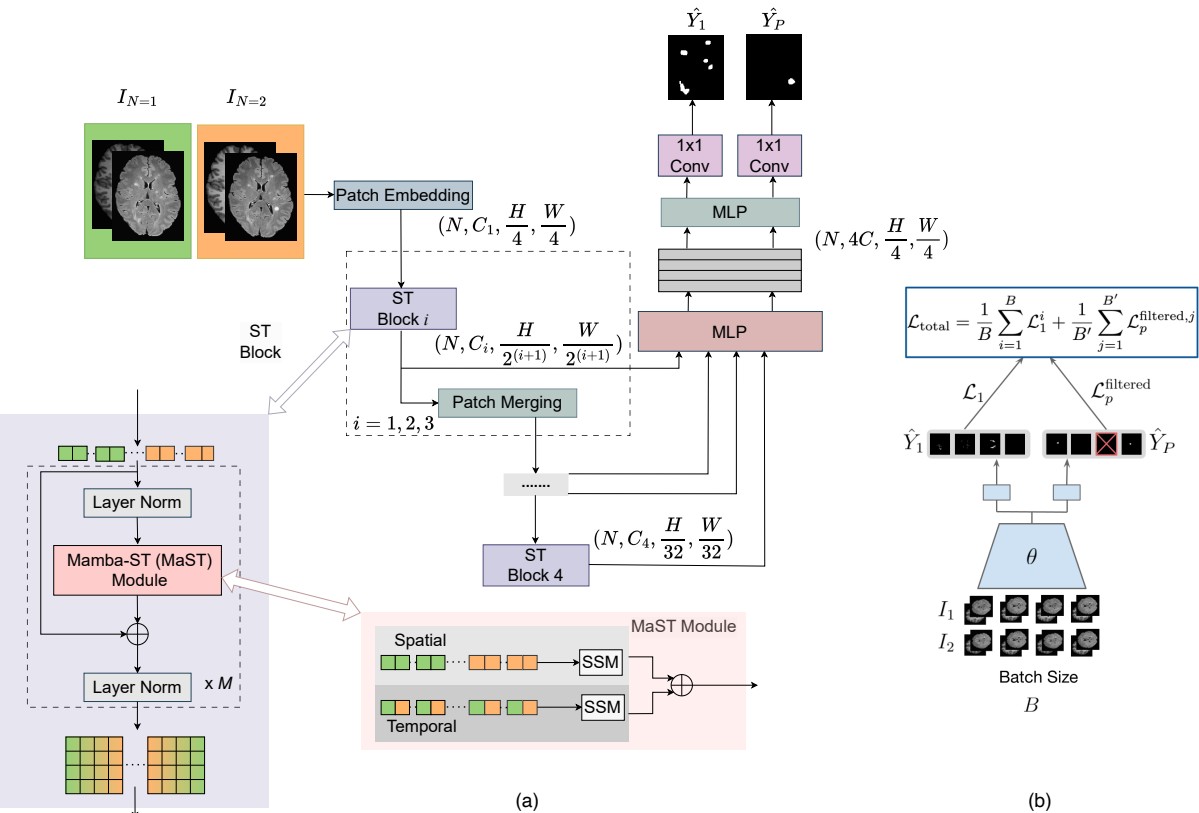

Figure 1: (a) *SegMaST* Architecture. Longitudinal image pairs are processed by a hierarchical encoder utilizing Spatio-Temporal (ST) blocks, where a *Mamba*-based module (MaST) efficiently captures spatial and temporal dependencies. Multi-scale features are aggregated by a *CNN*-based decoder, which feeds a dual-head module to predict baseline ($\hat{Y}_1$) and progression ($\hat{Y}_P$) masks. (b) *Imbalance-Aware Loss Accumulation.* The total loss combines the baseline term ($\mathcal{L}_1$) with a filtered progression term ($\mathcal{L}_p^{\text{filtered}}$). This mechanism selectively retains a balanced subset of zero-mask samples to prevent the high prevalence of non-progression from biasing the model.

The final loss is computed as the sum of mean terms, as shown in Figure 1 - (b), where $B$ is the original batch size and $B'$ is the number of samples used in the progression head after filtering. There is a trade-off between excluding all zero masks, which could bias training towards detecting only positive cases, and retaining all zero masks, which could lead to the model ignoring changes in progression. Our approach ensures a proportionate distribution, striking a balance between these extremes. The impact of different proportions is analyzed in an ablation study (section 4.1).

## 3. Experimental Setup

**Datasets.** We evaluate *SegMaST* on three datasets spanning two distinct brain pathologies: MS and diffuse glioma. The MS dataset was derived from a well-characterized subgroup of the cohort of our *in-house* observational MS study, TUM-MS (Bayas et al., 2024). This dataset contains MRI scans at baseline, 6, 12, and 24 months (3–4 time points per patient). Each subject has a baseline whole-lesion mask and new or enlarging lesion masks for follow-

ups. We use 3D FLAIR and T2 volumes of size $193 \times 229 \times 193$ voxels ($1 \times 1 \times 1$ mm$^3$) for binary segmentation of baseline lesions and progression, with 224 successive time-point pairs (125 without progression and 99 with progression). The test-set for the *in-house* MS dataset includes 45 patients, of whom 25 exhibit no progression.

The second MS dataset is the public *MSSEG-2* dataset, which provides only new-lesion annotations of follow-up scans. We use the official training split, which consists of two-time-point 3D FLAIR scans of size $193 \times 229 \times 193$ voxels ($1 \times 1 \times 1$ mm$^3$), comprising 40 timepoint pairs (11 without progression and 29 with progression). Follow-up scans were acquired 1–3 years after the baseline scan. We use 32 subjects for training and 8 subjects for testing. The test set corresponds to the official challenge validation split, with subjects without new lesions excluded according to the challenge protocol.

The third dataset, *UCSF-ALTPD* (Fields et al., 2024), consists of multimodal MRI scans from glioma patients with two consecutive follow-up time points. The preprocessing includes skull stripping (Isensee et al., 2019), N4 bias correction (Tustison et al., 2010), coregistration, and rigid SRI24 atlas registration (Rohlfing et al., 2010). Each case provides FLAIR, T1, contrast-enhanced T1 (T1-ce), and T2 volumes of size $240 \times 240 \times 155$ voxels ($1 \times 1 \times 1$ mm$^3$), along with tumor masks delineating enhancing tissue (ET), surrounding nonenhancing FLAIR hyperintensity (SNFH), nonenhancing tumor core (NETC), and resection cavity (RC). In our setup, the baseline head performs binary whole-tumor segmentation, while the progression head predicts a binary difference mask between the two time points. We randomly sample 200 patients for training and evaluation (88 without progression and 112 with progression). The test set consists of 25 patients, of whom 16 exhibit no progression.

**Training and Evaluation.** *SegMaST* adopts a 2.5D slice-based cross-view strategy. During training, the network is exposed to slices from all three orthogonal anatomical planes (axial, coronal, and sagittal), following established cross-view training paradigms in MS (Aslani et al., 2019). During inference, for each voxel, predictions are obtained from the three orthogonal slices intersecting that voxel. The probability outputs from the pixels in each view are averaged and thresholded to determine the final voxel-wise segmentation.

As baselines, we include both 2.5D and 3D spatiotemporal models. In the 2.5D setting, we adopt a *SegFormer* and a convolutional *DynUNet* (MONAI's *nnUNet*) baseline, both of which follow the same cross-view inference strategy as *SegMaST*, aggregating predictions from the three orthogonal planes at the voxel level. We further evaluate 3D models including *nnUNet* (Isensee et al., 2021), *DynUNet*, *SwinUNETR*, and *LongiUNet-DW* - extending a standard *UNet* with temporal depth-wise (TDW) blocks (Rokuss et al., 2024). On the *in-house* MS dataset, we additionally train two cross-sectional *DynUNet* models (*DynUNet (CS)*), with progression estimated via subtraction. For *MSSEG-2*, we further compare against established longitudinal and cross-sectional baselines, including *Coact-Seg*, *Neuropoly*, and *SNAC* (Wu et al., 2023). All baseline models for the *in-house* and *UCSF-ALTPD* datasets, except *nnUNet*, employ dual convolutional output heads to ensure architectural consistency with *SegMaST*. Since *nnUNet* does not natively support multiple ground-truth outputs, we adopt a multi-class formulation (background, baseline-only, follow-up-only, and overlap), where the overlap class denotes voxels labeled as lesions at

both timepoints. At inference, baseline and progression masks are reconstructed from the predicted labels to enable fair comparison with dual-head models.

To ensure consistent and fair evaluation, we standardize training, inference, and augmentation settings across models. We employ the weighted patch-sampling strategy of (Zhang et al., 2022) with crop sizes tailored to each dataset (*In-house*: 128; *MSSEG-2*: 80; *UCSF-ALTPD*: 160), which is applied during training. At inference, we perform patch-wise sliding-window prediction over the full volume using the same voxel resolution and patch sizes as in training, with a 50% overlap. For 2.5D models, slices are processed at the original in-plane resolution, and predictions from the three orthogonal planes are aggregated voxel-wise to obtain the final 3D segmentation. Predicted connected components below minimum volume thresholds ($27 \text{ mm}^3$ for MS and $50 \text{ mm}^3$ for UCSF-ALTPD) are suppressed following (Saluja, 2023). The 2.5D models are lightweight ($\sim$9M parameters for MS and $\sim$16M for diffuse glioma) and trained for 150-500 epochs with batch size 32 and early stopping, whereas 3D models are larger ($\sim$15M for MS and $\sim$25M for diffuse glioma) and trained for up to 1000 epochs with batch size 8. Standard geometric augmentations, including random flips and rotations, are applied. All models are optimized using AdamW (Loshchilov and Hutter, 2017) (weight decay 0.05, learning rate $10^{-4}$) with cosine learning rate decay (Loshchilov and Hutter, 2016) and a 40-epoch warm-up (Goyal et al., 2017). The proposed *imbalance-aware loss accumulation* is applied to all models throughout training. Our experiments are conducted on an NVIDIA RTX A6000 GPU. Note that *nnUNet* uses its default training, inference, and augmentation pipeline; thus, these settings are not applicable.

In line with recent validation recommendations (Maier-Hein et al., 2022), we use the Dice score to assess voxel-wise segmentation quality and the lesion-wise F1 score to evaluate lesion-level detection. We additionally report positive predictive value (PPV) for the MS datasets. To evaluate clinical utility, we compute the Disease Activity Assessment (DAA), a case-level clinical metric that assesses whether a follow-up scan of an MS patient is correctly classified as exhibiting disease progression or stability. A prediction is considered correct if progression is detected for true progression cases and no progression is predicted for stable cases, consistent with clinical monitoring guidelines for MS (Wattjes et al., 2021). Following the methodology established by (Commowick et al., 2018), we exclude tiny lesions (defined as less than $11 \text{ mm}^3$) from the F1 calculation to filter out noise and focus on clinically significant observations. We apply the Dice score to both baseline and progression regions, with progression performance further stratified into progression (P) and non-progression (NP) cases. For NP cases, we assign a strict Dice score of 1 when the predicted progression mask is empty and 0 otherwise, reflecting the clinical requirement of correctly identifying stable patients.

## 4. Results

For the *in-house* MS dataset, Table 1 reports test-set performance for baseline and progression prediction. For the baseline head, 3D *nnUNet* achieves the highest Dice and F1, likely due to its extensive data augmentation and pre/post-processing pipelines. While *SegMaST* remains competitive at baseline, it is the best-performing model for progression, achieving the highest P Dice (0.513) and lesion-wise F1 (0.688), outperforming all 2.5D and 3D baselines. Although 2.5D *DynUNet* attains strong NP performance, it underperforms *SegMaST*

Table 1: Baseline (B), progression (P) and no-progression (NP) metrics for the *in-house* MS dataset. Values are mean (SD). **Bold** denotes the best model and underline denotes the second-best. * and ** indicate statistical differences to *SegMaST* using a paired t-test with significance values <0.05 and <0.01, respectively. Disease Activity Assessment (DAA) shows the percentage of correctly classified follow-up scans. Training ratio is computed from wall-clock training time (in hours) per fold on a single GPU, normalized to *SegMaST*.

| Model | Dim | Dice ↑ | | | Lesion-F1 ↑ | | PPV ↑ | | DAA (%) ↑ | Train Ratio ↓ |
|---|---|---|---|---|---|---|---|---|---|---|
| | | B | P | NP | B | P | B | P | All | |
| DynUNet (CS) | 3D | 0.620 | 0.170** | 0.000** | 0.552 | 0.157** | 0.580** | 0.134** | 28.9% | 3.53 |
| | | (0.162) | (0.166) | (0.000) | (0.158) | (0.192) | (0.179) | (0.127) | | |
| LongiUNet-DW | 3D | 0.510** | 0.380** | 0.520 | 0.526 | 0.455 | 0.380** | 0.343* | **64.4%** | 3.58 |
| | | (0.175) | (0.256) | (0.500) | (0.186) | (0.368) | (0.172) | (0.277) | | |
| Swin UNETR | 3D | 0.657** | 0.371** | 0.520 | 0.621* | 0.440* | **0.800**** | 0.493 | 57.8% | 4.67 |
| | | (0.149) | (0.317) | (0.500) | (0.132) | (0.402) | (0.122) | (0.413) | | |
| nnUNet | 3D | **0.695**** | 0.482 | 0.280** | **0.717*** | 0.588 | 0.751** | 0.491 | 60.0% | 5.71 |
| | | (0.136) | (0.274) | (0.448) | (0.110) | (0.367) | (0.135) | (0.311) | | |
| DynUNet | 2.5D | 0.635** | 0.466 | **0.600** | 0.599* | 0.515* | 0.695** | **0.501** | **64.4%** | 1.67 |
| | | (0.141) | (0.331) | (0.490) | (0.143) | (0.420) | (0.182) | (0.372) | | |
| SegFormer | 2.5D | 0.533** | 0.396 | 0.360** | 0.451** | 0.528 | 0.534** | 0.389 | 55.6% | 1.33 |
| | | (0.180) | (0.290) | (0.480) | (0.151) | (0.355) | (0.217) | (0.314) | | |
| SegMaST (ours) | 2.5D | 0.617 | **0.513** | 0.560 | 0.567 | **0.688** | 0.648 | 0.494 | **64.4%** | **1.00** |
| | | (0.148) | (0.270) | (0.496) | (0.148) | (0.320) | (0.193) | (0.286) | | |

Table 2: Comparison of segmentation models on the *MSSEG-2* dataset, with progression subjects only. Values are mean (SD). **Bold** denotes best performance and underline denotes second-best.

| Model | Dim | Dice ↑ | Lesion-F1 ↑ | PPV ↑ |
|---|---|---|---|---|
| SNAC | 3D | 0.531 | 0.307 | *N/A* |
| SNAC (VNet) | 3D | 0.568 | 0.576 | *N/A* |
| Neuropoly | 3D | 0.563 | 0.175 | *N/A* |
| CoactSeg | 3D | **0.638** | 0.620 | 0.636 |
| DynUNet | 3D | 0.586 (0.238) | 0.622 (0.276) | 0.636 (0.325) |
| LongiUNet-DW | 3D | 0.616 (0.177) | 0.684 (0.229) | 0.569 (0.216) |
| SwinUNETR | 3D | 0.538 (0.280) | 0.557 (0.330) | **0.642** (0.355) |
| nnUNet | 3D | 0.615 (0.202) | 0.691 (0.215) | 0.585 (0.248) |
| DynUNet | 2.5D | 0.527 (0.222) | 0.535 (0.216) | 0.453 (0.255) |
| SegFormer | 2.5D | 0.532 (0.226) | 0.518 (0.308) | 0.621 (0.327) |
| SegMaST (ours) | 2.5D | 0.620 (0.201) | **0.723** (0.235) | 0.630 (0.281) |

on progression, highlighting the benefit of explicit spatiotemporal modeling. The subtractive *DynUNet (CS)* performs poorly in both P and NP cases. *SegMaST* achieves DAA comparable to top-performing baselines, matching *DynUNet* and *LongiUNet-DW* (64.4%). Importantly, *SegMaST* achieves superior longitudinal performance at the lowest training cost, converging $5.71\times$ faster than *nnUNet*. On *MSSEG-2*, *SegMaST* achieves the highest lesion-wise F1 (0.723), surpassing all prior methods, including the *nnUNet* (0.691). It also attains a competitive Dice score of 0.620, on par with top 3D models such as *CoactSeg*. Figure. 2 qualitatively demonstrates improved progression delineation on the *in-house* dataset (top) and *MSSEG-2* dataset (bottom), with reduced false positives and missed lesions. The mean test-set metrics for baseline whole-tumor segmentation and P/NP prediction on the *UCSF-ALTPD* dataset are shown in Table 3. For the baseline task, 3D *nnUNet* achieves the highest Dice score (0.861), while *SegMaST* remains highly competitive (0.844) and out-

Table 3: Comparison of segmentation models for baseline (B), progression (P), and no progression (NP) mask prediction in glioma cases. Values are mean (SD). **Bold** and underline denote best and second-best. * and ** indicate statistical differences to *SegMaST* using a paired t-test with significance values <0.05 and <0.01, respectively. Training ratio is computed from wall-clock training time (in hours) per fold on a single GPU, normalized to *SegMaST*.

| Model | Dim | Dice ↑ | | | Train Ratio ↓ |
|---|---|---|---|---|---|
| | | **B** | **P** | **NP** | |
| DynUNet | 3D | 0.846 (0.092) | **0.520** (0.096) | 0.125 (0.331)** | 2.60 |
| LongiUNet-DW | 3D | 0.836 (0.096) | 0.496 (0.141) | 0.000 (0.000)** | 2.73 |
| Swin UNETR | 3D | 0.784 (0.153)* | 0.449 (0.131) | 0.063 (0.242)** | 3.13 |
| nnUNet | 3D | **0.861** (0.117)* | 0.508 (0.095) | 0.063 (0.242)** | 3.97 |
| DynUNet | 2.5D | 0.839 (0.090) | 0.493 (0.151) | 0.563 (0.496) | 1.47 |
| SegFormer | 2.5D | 0.791 (0.126)* | 0.497 (0.128) | 0.500 (0.500)* | 1.13 |
| SegMaST (ours) | 2.5D | 0.844 (0.075) | 0.511 (0.132) | **0.750** (0.433) | **1.00** |

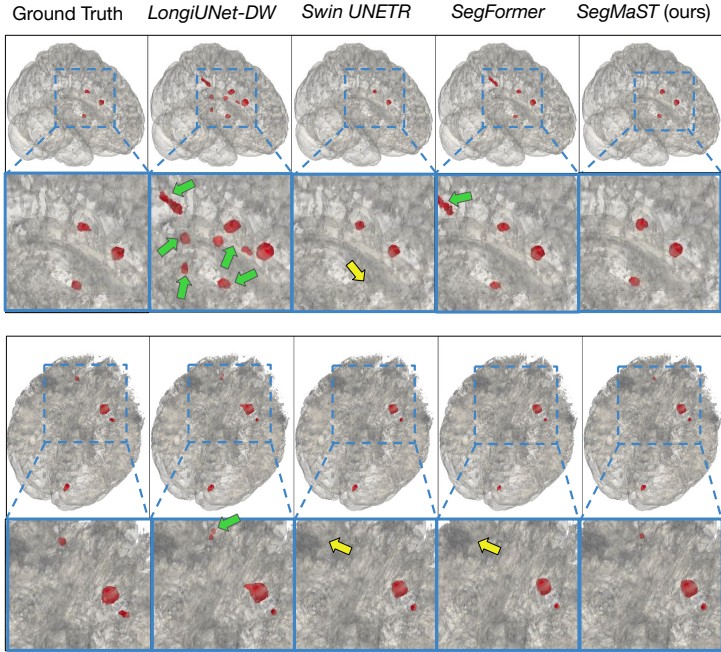

Figure 2: Exemplar results of *SegMaST* and other models for MS lesion progression segmentation. **Top:** *in-house* dataset. **Bottom:** *MSSEG-2* dataset. Green arrows indicate false positives, and yellow arrows indicate false negatives.

performs several 3D baselines. For the progression task, *SegMaST* provides the best overall performance, achieving the highest Dice for NP cases (0.750) while obtaining P Dice (0.511) comparable to *DynUNet* (0.520). Notably, the 3D baselines struggle significantly with NP cases, underscoring the difficulty of accurately modeling stable follow-up scans in these architectures. Furthermore, *SegMaST* maintains its efficiency advantage, training nearly 4× faster than *nnUNet*.

Overall, results across all experiments demonstrate that while *SegMaST* achieves performance comparable to state-of-the-art models such as *nnUNet* for baseline segmentation in

two of three datasets, it consistently outperforms all models in follow-up assessment across all three datasets, at a substantially lower computational cost.

## 4.1. Ablation Study - Contribution of No Progression Cases

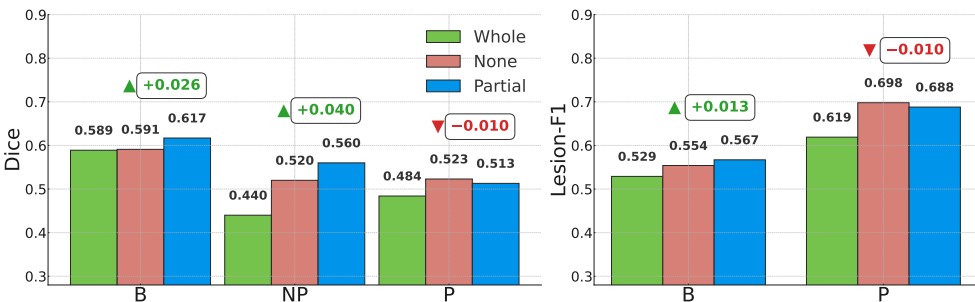

Figure 3: Comparison of three strategies for handling imbalance between progression (P) and no-progression (NP) cases. *Whole* uses all NP cases, *None* excludes them, and the proposed *Partial* strategy uses a balanced subset. Deviations indicate differences between *Partial* and *None*.

To counter the over-dominance of non-progression cases (which essentially are empty segmentation masks), we compare three distinct loss accumulation strategies: (1) **Whole:** all NP cases included, (2) **None:** all NP cases excluded, and (3) Our proposed **Partial:** or *imbalance-aware loss accumulation*, where a subset of NP cases is included, maintaining an equal number of NP and P cases per batch. As shown in Figure 3, the Whole strategy yields the lowest P Dice (0.484) and P F1 (0.619) due to the dominance of NP samples. The None setting achieves the highest P Dice (0.523) and P F1 (0.698) but performs poorly on B and NP. Our Partial strategy offers the best overall balance, obtaining the highest B Dice (0.617), NP Dice (0.560), and B F1 (0.567), while maintaining strong P F1 (0.688). This demonstrates that *imbalance-aware loss accumulation* effectively balances B, P, and NP performance, an important aspect for clinical use.

## 4.2. Ablation Study - Lesion Size-Wise Analysis

To evaluate the robustness of *SegMaST* across varying lesion volumes, we perform a size-wise analysis on the *in-house* MS dataset, reporting results for baseline lesions and progression lesions (Table 4). For the baseline segmentation (Left), *SegMaST* demonstrates superior performance across medium and large lesions, achieving the highest Lesion-F1 and Dice scores in both the Medium (Lesion-F1: 0.746, Dice: 0.424) and Large (Lesion-F1: 0.915, Dice: 0.617) bins. In the challenging progression lesion analysis (Right), *SegMaST* exhibits a dominant trend, achieving the highest Lesion-F1 scores across all three bins: Small (0.235), Medium (0.703), and Large (0.889). It also achieves the best Dice scores for both Medium (0.598) and Large (0.720) progression lesions. This strong performance, particularly in the difficult small-to-medium progression bins, validates the ability of our *Mamba*-based spatio-temporal modeling to effectively capture subtle changes indicative of disease progression across different lesion scales.

Table 4: Size-wise lesion analysis on the private MS dataset. **Left:** baseline lesion analysis. Lesion bins are defined using global tertile-based volume thresholds across the dataset ($1\,mm^3$ isotropic voxels): Small ($\leq 84$)$\,mm^3$, Medium (84–208)$\,mm^3$, Large ($> 208$)$\,mm^3$. **Right:** analysis of new lesions in follow-up scans. Lesion bins use volume thresholds: Small ($\leq 43$)$\,mm^3$, Medium (43–113)$\,mm^3$ and Large ($> 113$)$\,mm^3$.

| Model | Lesion-F1 ↑ | Dice ↑ | Model | Lesion-F1 ↑ | Dice ↑ |
|---|---|---|---|---|---|
| *Small* ($N = 394$) | | | *Small* ($N = 20$) | | |
| DynUNet | 0.176 | 0.076 | DynUNet | 0.034 | 0.010 |
| LongiUNet-DW | 0.186 | **0.216** | LongiNet-DW | 0.136 | 0.035 |
| Swin UNETR | **0.298** | 0.110 | Swin UNETR | 0.000 | 0.000 |
| SegFormer | 0.139 | 0.087 | SegFormer | 0.229 | **0.185** |
| SegMaST (ours) | 0.292 | 0.196 | SegMaST (ours) | **0.235** | 0.097 |
| *Medium* ($N = 388$) | | | *Medium* ($N = 21$) | | |
| DynUNet | 0.525 | 0.272 | DynUNet | 0.081 | 0.172 |
| LongiUNet-DW | 0.609 | 0.382 | LongiNet-DW | 0.421 | 0.450 |
| Swin UNETR | 0.699 | 0.345 | Swin UNETR | 0.376 | 0.309 |
| SegFormer | 0.625 | 0.302 | SegFormer | 0.415 | 0.460 |
| SegMaST (ours) | **0.746** | **0.424** | SegMaST (ours) | **0.703** | **0.598** |
| *Large* ($N = 392$) | | | *Large* ($N = 21$) | | |
| DynUNet | 0.840 | 0.525 | DynUNet | 0.293 | 0.175 |
| LongiUNet-DW | 0.850 | 0.523 | LongiNet-DW | 0.871 | 0.612 |
| Swin UNETR | 0.891 | 0.601 | Swin UNETR | 0.761 | 0.634 |
| SegFormer | 0.816 | 0.530 | SegFormer | 0.746 | 0.683 |
| SegMaST (ours) | **0.915** | **0.617** | SegMaST (ours) | **0.889** | **0.720** |

Figure 4: Multi-timepoint analysis of *SegMaST*. FU1 and FU2 denote the first and second follow-up scan. Follow-up scores are split into progression (P) and no-progression (NP). Values are mean (SD). Entropy maps show predictive uncertainty with ground-truth lesion contours in green. (a) Localized uncertainty at FU1 and diffused uncertainty at FU2. (b) Increased uncertainty around new lesion at FU2.

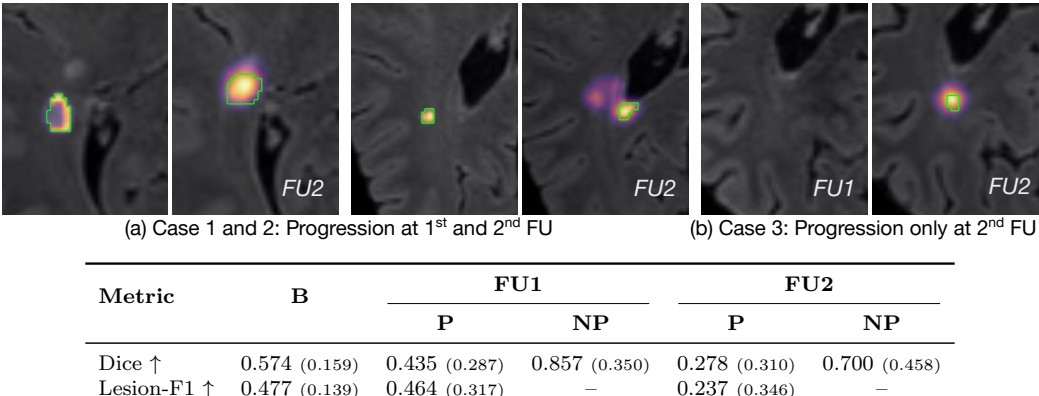

(a) Case 1 and 2: Progression at 1st and 2nd FU          (b) Case 3: Progression only at 2nd FU

| Metric | B | FU1 | | FU2 | |
|---|---|---|---|---|---|
| | | **P** | **NP** | **P** | **NP** |
| Dice ↑ | 0.574 (0.159) | 0.435 (0.287) | 0.857 (0.350) | 0.278 (0.310) | 0.700 (0.458) |
| Lesion-F1 ↑ | 0.477 (0.139) | 0.464 (0.317) | – | 0.237 (0.346) | – |

## 4.3. Ablation Study - Multi-Timepoint Spatio-Temporal Modeling

We investigate the scalability of *SegMaST* by extending the input from two to three consecutive timepoints using the *in-house* MS dataset. In this configuration, the model simultaneously predicts the baseline lesion mask (B) and two follow-up progression masks corresponding to the first (FU1) and second (FU2) follow-up scans. As shown in Table 4, the model maintains only slightly diminished baseline performance (Dice: 0.574) but exhibits a clear degradation in progression prediction as the temporal distance increases. Specifi-

cally, progression Dice decreases from 0.435 at FU1 to 0.278 at FU2, indicating increased uncertainty and temporal drift in long-term longitudinal modeling.

*Entropy maps* provide a mechanistic explanation for this temporal decay. At FU1, uncertainty remains spatially localized around lesion boundaries, whereas at FU2 it expands into a diffuse volumetric region, indicating that the network captures the correct anatomical neighborhood but loses sharp voxel-wise delineation (see Figure 4). This shift aligns with the observed Dice degradation. Overall, while *SegMaST* leverages longitudinal dependencies, spatial precision degrades over extended temporal windows. Several factors may contribute to this, such as (i) propagated registration inaccuracies, (ii) conflicting supervision signals, as the same image signal is a new lesion in FU1 and a stable lesion in FU2, or (iii) pronounced treatment changes and atrophy that accumulates over time.

## 5. Discussion & Conclusion

Longitudinal segmentation plays a crucial role in monitoring disease progression, yet existing methods often face challenges related to imbalanced datasets or foregoing the rich spatio-temporal information. To address these issues, we introduce *SegMaST*, a novel approach that jointly segments baseline and follow-up images while incorporating an *imbalance-aware loss accumulation* strategy to effectively manage the dominance of non-progression cases. Our results demonstrate that *SegMaST* achieves superior performance for longitudinal progression modeling compared to established baseline methods, including both convolutional and attention-based models, by leveraging rich spatio-temporal information in medical imaging data. Notably, this improvement is consistent across two distinct and challenging clinical scenarios: multiple small, newly appearing lesions in MS and continuous growth patterns in gliomas. These findings underscore the versatility and robustness of *SegMaST* for advancing longitudinal disease progression analysis.

An important limitation of our framework, and longitudinal image analysis in general, is the dependence on accurate image registration. While *SegMaST* exploits spatio-temporal dependencies to maintain consistency, it relies on good pre-alignment of baseline and follow-up scans. In clinical practice, perfect alignment is non-trivial; rigid registration fails to capture non-linear anatomical shifts, while deformable registration can introduce warping artifacts that obscure true pathological changes or create false progression. Consequently, residual misalignment can disrupt the spatial correspondences our spatio-temporal module relies on, potentially introducing noise that negatively impacts detection sensitivity and specificity. We explicitly designed *SegMaST* to be a disease-agnostic framework for longitudinal image analysis. Looking ahead, a promising direction for improving longitudinal assessment is the integration of clinical covariates alongside image data. Disease progression is often contextualized by non-imaging factors such as patient age, genetic markers, and specifically, treatment status. Incorporating these variables, for example via conditioning mechanisms within the network bottleneck, could provide the model with a prior regarding the likelihood of progression versus stability (e.g., distinguishing pseudo-progression from true recurrence in glioma). By moving beyond image changes alone and fusing multimodal clinical data, future iterations could offer a more holistic and clinically accurate evaluation of therapeutic response. We make our codes for *SegMaST* and the *imbalance-aware loss accumulation* publicly available at https://github.com/Aswathi-Varma/SegMaST.

## Acknowledgments

Aswathi Varma, Mark Mühlau, Benedikt Wiestler, and Daniel Rueckert are supported by the DFG as part of the SPP *Radiomics* (project number 428223038).

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

## 6. Appendix

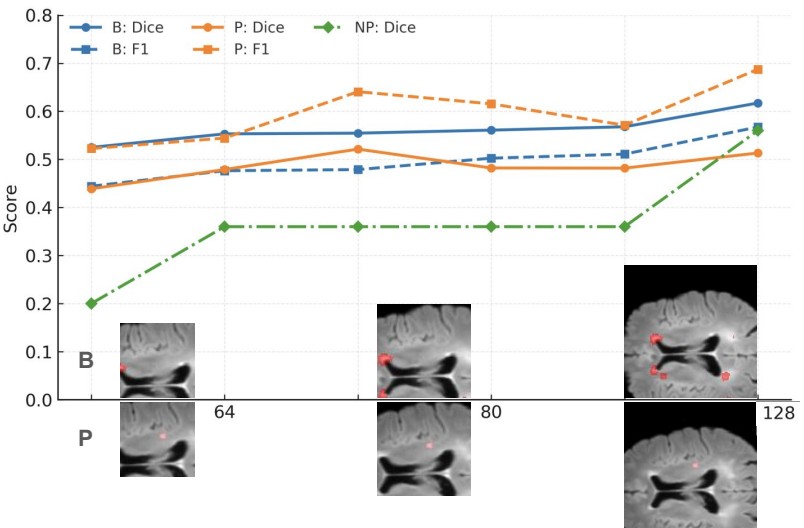

Figure 5: Ablation on patch size under weighted cropping. Larger patches ($40^2$ to $128^2$) provide more global context and improve Dice and F1 performance on the *in-house* dataset.

**Effect of Weighted Cropping Patch Size.** When applying weighted cropping on the in-house dataset, the results show that larger patch sizes consistently outperform smaller ones in terms of both baseline Dice and P F1 (Figure 5). Weighted cropping makes small patches (e.g., $40^2$) highly local and biased toward lesion-centered regions, limiting global contextual cues and resulting in lower baseline Dice (0.525) and weaker NP F1 (0.523). As patch size increases, the model benefits from more anatomical context, which improves its ability to distinguish true progression from stable regions. This is reflected in the steady rise in baseline Dice, reaching 0.617 at $128^2$, and in the pronounced improvement in \*\*non-empty progression F1, which peaks at 0.688 at the same size. Progression Dice follows a similar upward trend, increasing from 0.439 at $40^2$ to above 0.51 at $128^2$. Overall, combining weighted cropping with larger patches ($128^2$) provides the optimal balance: weighted

sampling focuses the model on lesion-relevant areas, while the larger patch size supplies the necessary global context, yielding the strongest Dice and F1 performance for progression detection.

