# OpenReview forum: "SegMaST: Mamba-based Spatio-Temporal Modeling to Improve Longitudinal Detection and Segmentation"
_MIDL.io/2026/Conference — MIDL 2026 Poster_

### Official Review · Reviewer_JvPR · 2025-12-28

**Confidence:** 5
**Preliminary Rating:** 3
**Final Rating:** 4

**Summary:**

The paper proposes SegMaST, a mamba based state-space model approach for longitudinal disease segmentation and detection. The linear complexity method is proposed as an alternative to the quadratic complexity attention-based methods.  The proposed model jointly predicts all lesions at baseline and new lesions in the follow-up unlike cross-sectional methods which predict all lesions or purely longitudinal methods which only predict progression. The paper also applies an imbalance aware loss to account for the rarity of disease progression. Experiments are conducted on multiple sclerosis datasets, including an in-house cohort and MSSEG-2, as well as a glioma dataset (UCSF-ALTPD), with comparisons against a range of CNN- and Transformer-based baselines, showing improved performance for progression detection. While the empirical results are strong and the application of state-space models is timely, several aspects of the methodology and evaluation, such as definition of clinical metrics, the training and evaluation protocols, and the evidence supporting efficiency claims, limit the strength of the paper’s clinical and methodological significance.

**Strengths:**

Strengths:
- The SegMaST approach for longitudinal disease segmentation with linear complexity is an exciting and potentially superior alternative to more complex attention-based methods.
- The joint formulation of baseline lesion segmentation and follow-up progression detection is conceptually sound and aligns with how longitudinal medical imaging data are annotated and analysed in practice.
- The paper presents a comprehensive evaluation against several relevant benchmarks.
- Ablation studies analysing the impact of inclusion of No-progression cases and the lesion size-wise analysis.
- The work demonstrates the potential of state-space models for longitudinal medical image analysis.
- The paper is generally well-written.

**Weaknesses:**

Weaknesses:

- The use of the term 2.5D is potentially misleading, as training appears to be purely 2D with multi-view aggregation applied only at inference. This differs from common usage in the literature, where 2.5D typically refers to stacked adjacent slices or cross-view inputs during training [1–3].
- Several key methodological details affecting reproducibility are missing or unclear, including the inference input resolution, the spatial dimensions of the volumes, and whether patch-based sampling is used only during training or also at inference.
- The DAA score is introduced without a clear formal definition or reference
- The phrasing around the evaluation strategy is ambiguous and does not clearly establish whether the same 2.5D inference strategy used for SegMaST is also applied to SegFormer.
- The training and evaluation protocol for the MSSEG-2 dataset is unclear. The paper should explicitly specify the number of patients used for training, validation, and testing.
- Since the proposed inference strategy could also be applied to standard 2D architectures (e.g. U-Net), including such baselines in the comparison would have strengthened the paper’s claims.
- While the paper motivates the use of state-space models through computational advantages over attention-based methods, no empirical results (e.g. runtime, memory usage) or quantitative discussion are provided to support these claims.
- Despite emphasising clinical relevance, the paper does not discuss model interpretability, explainability, or limitations, which are important considerations
- In line with MIDL’s open-science policy, the authors provide a link to their code; however, the repository is currently inaccessible (returns a 404 error).

[1] Zhang, H., Valcarcel, A. M., Bakshi, R., Chu, R., Bagnato, F., Shinohara, R. T., ... & Oguz, I. (2019, October). Multiple sclerosis lesion segmentation with tiramisu and 2.5 d stacked slices. In International Conference on Medical Image Computing and Computer-Assisted Intervention (pp. 338-346). Cham: Springer International Publishing.

[2] Zhang, Y., Liao, Q., Ding, L., & Zhang, J. (2022). Bridging 2D and 3D segmentation networks for computation-efficient volumetric medical image segmentation: An empirical study of 2.5 D solutions. Computerized Medical Imaging and Graphics, 99, 102088.

[3] Avesta, A., Hossain, S., Lin, M., Aboian, M., Krumholz, H. M., & Aneja, S. (2023). Comparing 3D, 2.5 D, and 2D approaches to brain image auto-segmentation. Bioengineering, 10(2), 181.

**Detailed Comments:**

Suggestions:
- Provide clarification on the input for 2.5D and 3D benchmarks.
- Clarify the training and inference setup and MSSEG-2 evaluation protocol.
- Justify the choice of baselines, in particular the omission of standard 2D models such as U-Net.
- Provide clarification or evidence for computational efficiency claims and briefly discuss interpretability and clinical considerations.
- The authors should temper their claims, especially in the conclusion. While the SegMaST achieves commendable performance, it does not consistently outperform the benchmarks. The key contribution is the remarkable performance at a fraction of the computation cost and the conclusion will benefit from being framed as such.

**Justification Of Final Rating:**

The authors’ rebuttal has adequately addressed most of the concerns raised in the initial review. While the work could still benefit from stronger baseline comparisons (for example, evaluating nnUNet in a 2D or 2.5D configuration), the inclusion of nnUNet in its 3D capacity and the addition of a comparable DynUNet baseline help contextualise the reported results. Furthermore, the revised manuscript reports the relative computational burden in terms of time-to-convergence, which strengthens the paper’s primary contribution around efficiency.

While there remains some scope for improvement, the revisions sufficiently address the main concerns, and the paper warrants an increase in rating by one point.

**Justification Of The Preliminary Rating:**

The paper addresses an important and clinically relevant problem and proposes a technically sound approach based on state-space models, with strong empirical results across multiple datasets. The idea of applying Mamba-style architectures to longitudinal medical image segmentation is timely and potentially valuable to the community. However, the paper as written leaves several methodological and evaluation details unclear, including aspects of the training and inference pipeline and the fairness of comparisons across baselines.

Additionally, the authors emphasise the computational benefits of the proposed method, but provide limited empirical evidence or discussion to substantiate these claims. While the results indicate that the proposed approach has strong potential, the current discussion does not adequately support or contextualise this aspect of the contribution.

While these issues do not undermine the overall promise of the approach, they limit confidence in the reproducibility and interpretability of the results.

**Questions To Address In The Rebuttal:**

The authors are encouraged to carefully address the points raised above, particularly those affecting clarity, reproducibility, and evaluation fairness, as doing so would significantly strengthen the contribution.

---

> ### Author Response · Authors · 2026-01-25
> **We clarify the evaluation protocol, add a new baseline and efficiency analysis, and expand interpretability and clinical discussion.**
>
> We are encouraged by the reviewer’s recognition of the *SegMaST* approach as a potentially superior alternative to attention-based methods, noting its linear complexity and the conceptual soundness of our joint baseline and progression formulation. We also appreciate the positive remarks regarding our evaluation, ablations, and manuscript clarity.
>
> ---
>
> ## **Addressing the comments**
>
> ### **1) Clarification of 2.5D training and inference**
>
> The reviewer notes that the term 2.5D may be misleading and that it is unclear whether the same inference strategy is applied to *SegFormer*. We have clarified this in the manuscript. We follow the cross-view 2.5D paradigm [1], where the network is trained on 2D slices from axial, coronal, and sagittal planes. At inference, voxel-wise predictions are generated by averaging probabilities from the three intersecting orthogonal slices. To ensure a fair evaluation, this same inference and aggregation strategy is applied consistently across all 2.5D baselines.
>
> [1] Aslani, Shahab, et al. "Multi-branch convolutional neural network for multiple sclerosis lesion segmentation." NeuroImage 196 (2019): 1-15.
>
> ---
>
> ### **2) Clarification on inference resolution, volume dimensions, and patch-based sampling**
>
> We revised the Training and Evaluation section to explicitly describe these details. All datasets are used at native isotropic spacing, with typical dimensions of $(193 \times 229 \times 193)$ voxels for MS datasets and $(240 \times 240 \times 155)$ voxels for diffuse glioma. Patch-based sampling with weighted cropping is used during training with dataset-specific patch sizes. During inference, patch-wise sliding-window prediction is performed at the same resolution and patch sizes with 50% overlap. For 2.5D models, inference is conducted at native in-plane resolution with voxel-wise aggregation from axial, coronal, and sagittal planes.
>
> ---
>
> ### **3) Definition of DAA score**
>
> The reviewer notes that the **Disease Activity Assessment** (DAA) score lacks a formal definition. We have added a formal definition and citation in the revised manuscript. DAA is a case-level clinical metric that evaluates whether a follow-up scan of an MS patient is correctly classified as exhibiting disease progression or stability. A prediction is considered correct if progression is detected for progression cases and no progression is predicted for stable cases, in line with MS monitoring guidelines.
>
> ---
>
> ### **4) MSSEG-2 training and evaluation protocol**
>
> The reviewer requests clarification of MSSEG-2 splits. We use the official MSSEG-2 split with **32 subjects for training** and **8 subjects for testing**. The test set corresponds to the official challenge validation split, as no separate held-out test labels are publicly available. These details are now explicitly reported in the *Datasets* section.
>
> ---
>
> ### **5) Missing standard 2D baseline**
>
> The reviewer suggests including a standard 2D baseline. We implemented a convolutional **2.5D *DynUNet*** baseline and report its performance in the revised manuscript. This model applies the same cross-plane aggregation and sliding-window inference as *SegMaST*, enabling a fair comparison between convolutional and transformer-based architectures. The results show that while convolutional models are precise in non-progression cases, *SegMaST* maintains superior sensitivity for detecting new lesions (**Progression Dice/F1**).
>
> ---
>
> ### **6) Computational efficiency evidence**
>
> The reviewer requests empirical runtime evidence. We added training-time comparisons across models, reported as relative training-time ratios with respect to *SegMaST*. These results provide empirical evidence of computational efficiency. We plan to include detailed runtime and memory profiling in future work.
>
> ---
>
> ### **7) Lack of Interpretability, Explainability, and Clinical Considerations**
>
> We have expanded the manuscript to include interpretability and limitations. We added multi-timepoint ablations demonstrating degraded spatial precision with increasing temporal gaps. We discuss limitations from registration errors, conflicting longitudinal supervision, and long-term anatomical changes due to treatment and atrophy. We further outline future directions, including integrating clinical covariates (e.g., age and treatment status) in the *Discussion* section.
>
> ---
>
> ### **8) Code repository accessibility**
>
> We sincerely apologize for this oversight. The repository is now public, and a persistent link and documentation will be provided in the camera-ready version.
>
> ---
>
> We added new experiments and revised the manuscript to clarify the evaluation protocol, experimental setup, and implementation details. Specifically, we introduced a convolutional 2.5D *DynUNet* baseline under identical inference settings and added training-time comparisons to quantify computational efficiency. These revisions provide a more rigorous validation of the *SegMaST* approach.

---

> > ### Comment · Reviewer_JvPR · 2026-01-29
> >
> > Thank you for the detailed rebuttal and added clarifications. Most of the raised concerns have been adequately addressed, and the addition of the nnUNet and DynUNet baselines further strengthens the paper’s contributions. However, the authors are encouraged to further temper the language in the conclusion, as the claim of “superior” performance is not fully supported by the evidence presented in the table. The authors now explicitly identify the relative computational burden of the baseline methods, and framing the contribution around this efficiency–performance trade-off would further strengthen the paper.
> >
> > Given the comprehensive rebuttal, I am happy to raise my rating by one point, while encouraging the authors to meaningfully adapt their claims in the camera-ready version.

---

> > > ### Author Response · Authors · 2026-01-30
> > >
> > > Thank you for the constructive feedback and for increasing the rating, we really appreciate it. We will refine the language in the conclusion section and further frame our contributions around the efficiency performance trade-off in the camera ready version.

---

> ### Author Response · Authors · 2026-01-29
> **Follow-up on revisions and additional experiments**
>
> Thank you again for your detailed and constructive feedback. We have carefully revised the manuscript to address all points raised, including clarifying the 2.5D training and inference protocol, dataset splits, additional baselines, computational efficiency and code availability.
>
> We would greatly appreciate it if you could let us know whether these revisions sufficiently address your concerns, or if any aspects would benefit from further clarification.

---

### Official Review · Reviewer_CpuJ · 2026-01-09

**Confidence:** 5
**Preliminary Rating:** 4
**Final Rating:** 4

**Summary:**

The paper presents SegMaST, a spatio-temporal framework based on the Mamba state-space model for longitudinal medical image analysis. The model is designed to jointly segment baseline images and localize progression (new or enlarging lesions) in follow-up scans. To address the prevalence of stable cases in clinical data, the authors introduce an imbalance-aware loss accumulation strategy. Evaluation is conducted on Multiple Sclerosis (MS) and diffuse glioma datasets, demonstrating improvements in lesion detection and segmentation consistency over standard CNN and attention-based baselines.

**Strengths:**

1. Investigating longitudinal disease progression is fundamental for quantifying treatment efficacy and disease evolution, yet it remains significantly more challenging than cross-sectional analysis.
2. The use of a Mamba-based selective state-space model (SSM) addresses the quadratic complexity  of standard attention, allowing for global receptive fields with linear complexity.
3. The introduction of a filtering mechanism to balance "progression" and "no-progression" samples in training batches is a well-motivated response to the skewed distribution of real-world clinical datasets.

**Weaknesses:**

1. While the joint modeling of two timepoints is technically sound, it relies on a specific "paired input" requirement that can be difficult to satisfy in standard clinical practice. In many settings, perfectly registered, high-quality longitudinal pairs are rare or difficult to obtain due to varying acquisition protocols and patient follow-up inconsistency. The model's utility may be limited by this high bar for input data.
2. In the UCSF-ALTPD glioma dataset, progression is not merely a biological spatio-temporal process; it is heavily influenced by clinical interventions such as surgery (resection cavities), radiation, and chemotherapy. The current task setting treats these changes as purely visual cues. Similar to the BraTS challenge, which distinguishes between pre-treatment and post-treatment segmentation tasks, the authors should have addressed how treatment effects confound the detection of "true" disease progression.
3. Disease progression is a multi-factor phenomenon. The current model relies solely on image features , ignoring demographic data (age, gender) or treatment status. Without these non-imaging covariates, the model lacks the necessary context to reliably distinguish between aggressive progression and treatment-induced pseudoprogression, a critical distinction in neurological care.
4. The technical novelty of this work appears limited. Although the authors incorporate the Mamba architecture and an imbalance-aware loss accumulation strategy, these changes remain largely incremental. Moreover, recent studies have leveraged VLM–based or diffusion-based segmentation paradigms and have reported more substantial gains in segmentation performance.

**Detailed Comments:**

See Weaknesses.

**Justification Of Final Rating:**

While the paper still presents considerable room for improvement, I strongly encourage the authors to genuinely pursue the directions outlined in their future work. However, given the authors' honest acknowledgment of the current limitations and their conscientious response during the rebuttal, I am willing to increase my rating.

**Justification Of The Preliminary Rating:**

While the problem is important and timely, the current task formulation appears somewhat idealized relative to the complexities of real-world oncology and neurology. Consequently, the technical novelty also leaves substantial room for improvement.

**Questions To Address In The Rebuttal:**

1. Regarding the UCSF-ALTPD dataset, did the authors consider the presence of resection cavities or post-surgical changes as a specific class or confounding variable?
2. Could the authors elaborate on how the model could incorporate clinical metadata (e.g., time since last treatment) to improve progression detection?
3. How does the performance compare if the model is extended to handle more than two timepoints, which is common in chronic MS monitoring?

---

> ### Author Response · Authors · 2026-01-25
> **We address clinical feasibility, dataset confounds and add multi-timepoint longitudinal evaluation.**
>
> We thank the reviewer for the constructive feedback and appreciate the recognition of our longitudinal modeling approach, Mamba-based state-space design, and imbalance-aware filtering strategy. We address the comments below and revised the manuscript accordingly.
>
> ---
>
> ### **1. Paired Longitudinal Input Requirement and Clinical Feasibility**
>
> Thank you for raising this practical concern regarding real-world applicability. We agree that longitudinal image processing faces challenges regarding data consistency and registration; however, we emphasize two key points. *Clinical Relevance* – Longitudinal imaging is not merely "available" but is the standard of care for the pathologies addressed in this work (MS and oncology). In these domains, patient management relies almost exclusively on comparative follow-up assessments. Therefore, developing methods that address the temporal dimension is crucial for clinical translation. *Robustness and Availability* – while perfect registration is challenging, modern preprocessing pipelines are increasingly robust to imperfect alignment. We acknowledge this as an important future research direction and revised the *Discussion* accordingly.
>
> ---
>
> ### **2. Treatment-Induced Confounds in the UCSF-ALTPD Glioma Dataset**
>
> We agree that treatment effects (eg., resection cavities, radiation necrosis) present complex visual patterns. However, the clinical gold standard, the *RANO 2.0 criteria [1]*, explicitly prioritize measuring volumetric changes in contrast-enhancing and FLAIR-hyperintense areas as the primary evidence for response assessment. Etiology (treatment effect vs. tumor) is a secondary clinical determination. *SegMaST* is designed to automate this primary volumetric quantification, providing the objective measurements required for downstream clinical decision-making.
>
> [1] Wen PY, van den Bent M, Youssef G, et al. RANO 2.0: Update to the Response Assessment in Neuro-Oncology Criteria for High- and Low-Grade Gliomas in Adults. J Clin Oncol. 2023;41(33):5187-5199.
>
> ---
>
> ### **3. Absence of Non-Imaging Clinical Covariates**
>
> While we agree that incorporating metadata (e.g., MGMT status, treatment protocols) could boost performance for specific diagnostic tasks, we deliberately designed *SegMaST* as a modality-agnostic, image-based framework. Our goal was to create a generalized method capable of capturing spatio-temporal dynamics across different pathologies, as demonstrated by our experiments on MS (multiple small lesions) and gliomas (large, complex tumors). Hard-coding disease-specific metadata into the architecture would limit this generalizability and increase dependency on curated clinical databases, which are often incomplete. However, we agree that this is a valuable direction for disease-specific optimization and have added a discussion on how users could fuse metadata into the SegMaST architecture for downstream clinical applications.
>
> ---
>
> ### **4. Technical Novelty**
>
> We acknowledge VLM- and diffusion-based models as promising. Our work focuses on supervised longitudinal progression segmentation under extreme class imbalance and temporal dependency. We introduce a state-space spatio-temporal framework with progression supervision, imbalance-aware loss, and multi-timepoint uncertainty analysis, addressing challenges not targeted by current diffusion or VLM approaches. Exploring foundation-model or diffusion-based longitudinal modeling is an exciting direction for future work.
>
> ---
>
> ### **5. Handling Post-Surgical Changes in UCSF-ALTPD**
>
> As discussed above, we explicitly designed *SegMaST* as an image-based framework and did not model treatment effects. Consequently, and following *RANO 2.0* guidelines, we segmented contrast-enhancing and FLAIR-hyperintense tumor parts and their change during baseline and follow-up, respectively.
>
> ### **6. Extension Beyond Two Timepoints**
>
> We thank the reviewer for highlighting the importance of multi-timepoint longitudinal evaluation. To address this, we added a spatio-temporal ablation study extending *SegMaST* to three consecutive timepoints on the *in-house* MS dataset, jointly predicting the baseline and two follow-up masks. We observe stable baseline segmentation but progressive degradation in progression prediction with increasing temporal distance. We also provide qualitative uncertainty visualizations using entropy maps, showing boundary-localized uncertainty at early follow-ups that becomes more diffuse over time, indicating temporal drift and reduced voxel-level precision. This analysis demonstrates limitations of long-horizon longitudinal progression modeling.
>
> ---
>
> In summary, we clarified clinical feasibility and dataset considerations, justified the image-based and disease-agnostic design of *SegMaST*, and addressed concerns regarding technical novelty and treatment-induced confounds. We further extended the framework to multi-timepoint longitudinal modeling, highlighting both its capabilities and limitations.

---

> > ### Comment · Reviewer_CpuJ · 2026-01-25
> >
> > While the paper still presents considerable room for improvement, I strongly encourage the authors to genuinely pursue the directions outlined in their future work. However, given the authors' honest acknowledgment of the current limitations and their conscientious response during the rebuttal, I am willing to increase my rating by 1 point.

---

> > > ### Author Response · Authors · 2026-01-26
> > >
> > > Thank you for your constructive feedback and for increasing the rating. We are glad that our rebuttal and clarifications were well received. We greatly appreciate your encouragement and will prioritize the suggested directions in our future work.

---

### Official Review · Reviewer_rw8y · 2026-01-10

**Confidence:** 4
**Preliminary Rating:** 2
**Final Rating:** 4

**Summary:**

The paper proposed a Mamba-based framework for segmentation and lesion detection. They evaluated their model on two Brain datasets with patients from the Multiple Sclerosis (MS) and glioma cohorts. Their comparison with some of the existing baseline models shows superior performance in most cases but poor in some cases.

**Strengths:**

1. Evaluated on 2 datasets and compared with multiple baseline models.
2. It has some technical novelty and employs a suitable loss function to handle data imbalance.
3. The description is clear and easy to follow.

**Weaknesses:**

1. This paper didn't compare their segmentation performance with nnUnet which is mostly used as a baseline.
2. The generalizability of this method is questionable.
3. The performance doesn't always improve.
4. Explanation of poor performance (in some cases) is limited.

**Detailed Comments:**

1. For medical image segmentation, nnUnet is considered one of the most popular baselines because it tends to get good performance over any medical data. However, this paper doesn't include this nnUnet model as a baseline, which questions the credibility of the proposed model in performance.

2. The title "SegMaST: Mamba-based Spatio-Temporal Modeling to Improve Longitudinal Disease Detection and Segmentation" is written in a way to make it sound generalizable for different medical data (e.g., Brain, Kidney, Prostate, etc). However, their evaluation only contains Brain datasets. So, it would be better to change the title to reflect the use case.

**Justification Of Final Rating:**

The authors have addressed my comments. They compared their model with the nnUnet model. Also, they expanded the Discussion section to explicitly analyze limitations of longitudinal segmentation. And finally, clarified the scope and generalizability of their model.

**Justification Of The Preliminary Rating:**

As the paper doesn't compare their model's performance with a strong baseline like nnUnet, the contribution is questionable. I would definitely change my rating if I see improvement over the nnUnet model because this model is a very strong baseline and is used as a standard model.

**Questions To Address In The Rebuttal:**

Please refer to the detailed comments and weaknesses section and address them.

---

> ### Author Response · Authors · 2026-01-25
> **We added nnUNet comparisons, clarified the generalizability, and expanded performance and limitation analysis.**
>
> We thank the reviewer for the constructive feedback and recognition of our multi-dataset evaluation, technical novelty, and imbalance-aware loss formulation. We address the concerns below.
>
> ---
>
> ### **1. Missing nnU-Net Baseline Comparison**
>
> The reviewer notes that our paper does not include *nnUNet* as a baseline, which is widely considered a strong default for medical image segmentation. We thank the reviewer for this suggestion and have now included *nnUNet* (3D) as a baseline across all three datasets (see Tables 1–3).
>
> Our updated results demonstrate that:
>
> 1. **Baseline Segmentation:** *SegMaST* is comparable to *nnUNet* in 2 out of 3 datasets.
> 2. **Follow-up/Progression Assessment:** *SegMaST* outperforms *nnUNet* across all three datasets.
> 3. **Efficiency:** *SegMaST* achieves this superior longitudinal accuracy at a fraction of the cost, converging up to 5.71$\times$ faster than *nnUNet*.
>
> **Clarification on Comparison Fairness:** *nnUNet* is a comprehensive framework that incorporates extensive data augmentation, optimized pre- and post-processing, and automated hyperparameter tuning. While these confounding factors typically boost baseline scores, they do not necessarily resolve the spatiotemporal challenges inherent in progression prediction. By outperforming this highly-optimized framework in follow-up tasks, while using a significantly more efficient architecture, we believe the results further highlight the effectiveness of *SegMaST*’s specialized spatiotemporal design.
>
> ---
>
> ### **2. Questionable Generalizability and Title Scope**
>
> We appreciate this observation regarding the scope of our validation. While our current expertise as a neuroradiology research group has naturally led us to prioritize brain-specific datasets for this study, we have intentionally designed *SegMaST* as a disease-agnostic temporal framework. *SegMaST* jointly processes multiple timepoints to model longitudinal dependencies and predict baseline and follow-up masks, explicitly localizing new pathological regions. We further introduce an imbalance-aware loss accumulation strategy to address class imbalance in clinical cohorts. Importantly, these components in *SegMaST* are modality- and organ-agnostic and do not rely on brain-specific priors, making the framework directly applicable to other longitudinal imaging scenarios (e.g., lung CT or abdominal MRI). We therefore believe the current title accurately reflects the methodological scope of the work rather than the specific datasets evaluated. We have clarified this scope in the Discussion section. Evaluating *SegMaST* on additional organs and modalities is an interesting direction for future work.
>
> ---
>
> ### **3. Performance Does Not Consistently Improve**
>
> We appreciate the opportunity to clarify the performance profile of *SegMaST*. Our results show that *SegMaST* is a specialized architecture prioritizing longitudinal accuracy and computational efficiency.
>
> 1. **Superiority in Progression (Follow-Up) Tasks:** Baseline segmentation is a mature task where many models (including *nnUNet*) perform well, but the primary goal of this work is progression detection. *SegMaST* consistently outperforms all baselines, including *nnUNet* and *DynUNet*, in follow-up assessment across all three datasets, highlighting the advantage of explicit spatiotemporal modeling.
>
> 2. **Comparable Baseline Performance:** *SegMaST* remains comparable to *nnUNet* at baseline in 2 of 3 datasets. Matching a highly optimized framework like *nnUNet* at baseline while outperforming it on progression tasks indicates strong suitability for longitudinal applications.
>
> 3. **Efficiency and Resource Footprint:** *SegMaST* achieves this performance at a fraction of the computational cost, converging up to 5.71$\times$ faster than competitive 3D baselines. This enables superior longitudinal accuracy with substantially lower training and inference overhead.
>
> ---
>
> ### **4. Limited Explanation of Poor Performance Cases**
>
> We thank the reviewer for this important point. We have expanded the *Discussion* section to explicitly analyze limitations of longitudinal segmentation. In particular, we discuss the dependence on accurate longitudinal image registration, noting that residual misalignment (from rigid or deformable registration) can introduce spurious temporal inconsistencies and degrade detection sensitivity and specificity. We further highlight that non-linear anatomical shifts, warping artifacts, and imperfect spatial correspondence can disrupt the spatio-temporal consistency assumptions of *SegMaST*.
>
> ---
>
> In summary, we added nnUNet comparisons, clarified the scope and generalizability of *SegMaST*, and provided an analysis of its performance profile. We show that *SegMaST* achieves competitive baseline segmentation, consistently superior longitudinal progression performance, and substantially improved computational efficiency. We further expanded the Discussion to clarify key limitations.

---

> > ### Comment · Reviewer_rw8y · 2026-02-02
> >
> > Thank you for addressing my comments. I am increasing my rating.

---

> ### Author Response · Authors · 2026-01-29
> **Follow-up on revisions and additional experiments**
>
> Thank you again for your thoughtful and detailed comments. We have revised the manuscript to address all concerns, including adding nnUNet as a baseline, clarifying the scope and generalizability of SegMaST and expanding the discussion on limitations.
>
> We would be grateful for your feedback on whether the revisions adequately address your concerns, or if additional clarification would be helpful.

---

> ### Comment · Area_Chair_3YQy · 2026-01-30
>
> Dear Reviewer,
>
> Please read the authors' response and post a comment indicating if your concerns have been addressed. Please also ensure your final rating is updated accordingly.
>
> Thank you.

---

### Author Response · Authors · 2026-01-25
**Summary of Revisions and Additional Experiments**

We thank the reviewers and Area Chairs for their careful evaluation and constructive feedback. We are encouraged by the reviewers’ recognition that “the paper presents a comprehensive evaluation against several relevant benchmarks” [R3], that the method is “conceptually sound and aligns with how longitudinal medical imaging data are annotated and analysed in practice” [R3], and that “the SegMaST approach for longitudinal disease segmentation with linear complexity is an exciting and potentially superior alternative to more complex attention-based methods” [R3]. Reviewers further highlighted that “the use of a Mamba-based selective state-space model addresses the quadratic complexity of standard attention” [R2], that the imbalance-aware training strategy is “a well-motivated response to the skewed distribution of real-world clinical datasets” [R2], and that the manuscript is “clear and easy to follow” [R1]. In response to the reviews, we have significantly strengthened the manuscript with new experiments, including a standardized nnUNet baseline, computational efficiency benchmarks, and a longitudinal ablation study.

---

### **1. Performance and Efficiency Benchmarking (R1, R3)**

To ensure a rigorous comparison, we integrated the 3D *nnUNet* and 2.5D *DynUNet* into our evaluation across all three datasets.

* **Superior Progression Detection:** While *nnUNet* is a formidable baseline for static segmentation, *SegMaST* consistently outperforms it in follow-up progression assessment (P and NP metrics), confirming that our spatiotemporal design captures temporal evolution more effectively than standard 3D frameworks.
* **Computational Advantage:** *SegMaST* achieves this superior accuracy while remaining highly efficient, converging up to $5.71\times$ faster than *nnUNet*.
* **Performance Stability:** *SegMaST* maintains performance comparable to these highly optimized frameworks in baseline tasks, demonstrating that it gains temporal sensitivity without sacrificing static segmentation accuracy.

---

### **2. Technical Rigor and Reproducibility (R3)**

We standardized our evaluation protocol to ensure transparency and fairness:

* **Unified Inference:** We clarified our 2.5D cross-view paradigm (axial, coronal, and sagittal aggregation). The identical aggregation strategy was applied to all 2.5D baselines, including *SegFormer* and *DynUNet*, to ensure fair comparison.
* **Methodological Transparency:** We now explicitly report input resolution, patch-based sampling, and inference configurations to ensure full reproducibility.
* **Open Science:** The code repository is now publicly accessible and verified, resolving the previous link error.

---

### **3. Longitudinal Depth and Interpretability (R2, R3)**

We extended our analysis to address the complexities of long-term disease monitoring:

* **Multi-timepoint Ablation:** We added an experiment extending *SegMaST* to three consecutive timepoints. Results show stable baseline performance but natural degradation in progression precision with increasing temporal distance - a key insight for long-horizon monitoring.
* **Explainability:** We provide qualitative uncertainty visualizations (entropy maps), showing uncertainty localized at lesion boundaries early and becoming more diffuse over time, aiding clinical interpretation of temporal drift.

---

### **4. Clinical Context and Generalizability (R1, R2)**

* **Real-world Utility:** While longitudinal modeling requires paired inputs, this is the standard of care for MS and Glioma. *SegMaST* automates volumetric quantification required by clinical guidelines such as RANO 2.0.
* **Robust Framework:** We deliberately maintained an image-based, modality-agnostic design to ensure *SegMaST* remains generalizable across diverse pathologies and centers without relying on often-incomplete clinical metadata (e.g., treatment history or demographics).

We believe these additions provide a comprehensive and transparent validation of *SegMaST* and clarify its methodological scope and clinical relevance. We hope this work contributes to ongoing efforts within the MIDL community toward robust and clinically meaningful longitudinal modeling.

---

### Author Rebuttal · Authors · 2026-01-25

**Rebuttal:**

We significantly strengthened the paper by expanding baselines, clarifying methodology, and improving clinical and technical discussion. **(1)** We added nnUNet (3D) across all datasets, showing SegMaST is comparable for baseline segmentation and consistently superior for longitudinal progression detection, while converging up to 5.71× faster, demonstrating strong efficiency advantages. **(2)** We clarified the scope and generalizability claims, explicitly discussing current brain-focused validation, title interpretation, and future extensions to other organs and modalities. **(3)** We added interpretability analyses and limitations related to registration errors, treatment effects, and long-term temporal drift.

We also improved methodological transparency and reproducibility. **(4)** We clarified the 2.5D training and inference protocol, ensured consistent aggregation across all 2.5D baselines, and detailed spatial resolution, patch sampling, and sliding-window inference. **(5)** We added a new 2.5D convolutional baseline, the DynUNet. **(6)** We introduced a multi-timepoint ablation study with uncertainty visualization to analyze temporal degradation and long-horizon challenges. Overall, these revisions improve rigor, clarity, clinical relevance, and validation, while reinforcing SegMaST’s contribution as an efficient spatiotemporal framework for longitudinal disease modeling.

**Supporting Material:**

/attachment/17b6f94fcde4bc2871d5145064c7017a69fa058b.pdf

---

### Meta-Review · Area_Chair_3YQy · 2026-02-07

**Recommendation:** Accept (Poster)
**Confidence:** 5

**Metareview:**

The reviewers were all positive about the paper and agreed that the topic of longitudinal disease progression detection is important and more challenging than standard standalone detection work. The idea of balancing progression and non‑progression samples is interesting and well-motivated. The main concerns were the missing nnUNet baseline and the need for a clearer discussion of limitations and generalizability. The authors addressed these points well in the rebuttal by adding the nnUNet‑3D comparisons, clarifying the 2.5D setup, adding an extra baseline and ablations, and expanding the discussion on scope and limitations. With these updates, I find that most of the reviewers' concerns were addressed, and I agree with their overall acceptance decision. I recommend accepting the paper as a poster.

---

### Decision · Program_Chairs · 2026-02-13

Accept (Poster)